# Vitamin K in COVID-19—Potential Anti-COVID-19 Properties of Fermented Milk Fortified with Bee Honey as a Natural Source of Vitamin K and Probiotics

**Amira Mohammed Ali** [1,2,*], **Hiroshi Kunugi** [3,4], **Hend A. Abdelmageed** [5,6], **Ahmed S. Mandour** [7,8], **Mostafa Elsayed Ahmed** [9,10], **Saboor Ahmad** [11] **and Amin Omar Hendawy** [12,13]

1   Department of Behavioral Medicine, National Institute of Mental Health, National Center of Neurology and Psychiatry, 4-1-1, Ogawahigashi, Kodaira, Tokyo 187-8553, Japan
2   Department of Psychiatric Nursing and Mental Health, Faculty of Nursing, Alexandria University, Smouha, Alexandria 21527, Egypt
3   Department of Psychiatry, Teikyo University School of Medicine, 2-11-1 Kaga, Itabashi-ku, Tokyo 173-8605, Japan; hkunugi@med.teikyo-u.ac.jp
4   National Center of Neurology and Psychiatry, Department of Mental Disorder Research, National Institute of Neuroscience, 4-1-1, Ogawahigashi, Kodaira, Tokyo 187-8502, Japan
5   Agriculture Research Center, Department of Bacteriology, Animal Health Research Institute, Ismailia Lab, First District, Ismailia 41511, Egypt; hendabdelmageed312@gmail.com
6   Laboratory of Veterinary Microbiology, Cooperative Department of Veterinary Medicine, Faculty of Agriculture, Tokyo University of Agriculture and Technology, 3-5-8 Saiwai-cho, Fuchu-shi, Tokyo 183-8509, Japan
7   Department of Animal Medicine (Internal Medicine), Faculty of Veterinary Medicine, Suez Canal University, Ring Road, Ismailia 41522, Egypt; dr_mandour@vet.suez.edu.eg
8   Department of Veterinary Medicine, Tokyo University of Agriculture and Technology, Tokyo 183-8509, Japan
9   Department of Plant Protection, Faculty of Agriculture, Damanhour University, Damanhour 22516, Egypt; xiphoidman2000@yahoo.com
10  Institute of Apiculture Research, Chinese Academy of Agricultural Science, Beijing 100081, China
11  Key Laboratory of Pollinating Insect Biology, Ministry of Agriculture, Institute of Apicultural Research, Chinese Academy of Agricultural Sciences, Beijing 100081, China; 2018y90100172@caas.cn
12  Department of Animal and Poultry Production, Faculty of Agriculture, Damanhour University, Damanhour 22516, Egypt; amin.hendawy@gmail.com
13  Department of Biological Production, Tokyo University of Agriculture and Technology, Tokyo 183-8509, Japan
*   Correspondence: mercy.ofheaven2000@gmail.com; Tel.: +81-042-346-1714

**Abstract:** Vitamin K deficiency is evident in severe and fatal COVID-19 patients. It is associated with the cytokine storm, thrombotic complications, multiple organ damage, and high mortality, suggesting a key role of vitamin K in the pathology of COVID-19. To support this view, we summarized findings reported from machine learning studies, molecular simulation, and human studies on the association between vitamin K and SARS-CoV-2. We also investigated the literature for the association between vitamin K antagonists (VKA) and the prognosis of COVID-19. In addition, we speculated that fermented milk fortified with bee honey as a natural source of vitamin K and probiotics may protect against COVID-19 and its severity. The results reported by several studies emphasize vitamin K deficiency in COVID-19 and related complications. However, the literature on the role of VKA and other oral anticoagulants in COVID-19 is controversial: some studies report reductions in (intensive care unit admission, mechanical ventilation, and mortality), others report no effect on mortality, while some studies report higher mortality among patients on chronic oral anticoagulants, including VKA. Supplementing fermented milk with honey increases milk peptides, bacterial vitamin K production, and compounds that act as potent antioxidants: phenols, sulforaphane, and metabolites of lactobacilli. Lactobacilli are probiotic bacteria that are suggested to interfere with various aspects of COVID-19 infection ranging from receptor binding to metabolic pathways involved in disease prognosis. Thus, fermented milk that contains natural honey may be a dietary manipulation capable of correcting nutritional and immune deficiencies that predispose to and aggravate COVID-19. Empirical studies are warranted to investigate the benefits of these compounds.

**Keywords:** vitamin K; phylloquinone; menaquinones; Coronavirus disease 2019; COVID-19; fermented milk; yogurt; dairy products; milk peptides; honey; probiotics; in silico; in vitro; vitamin K antagonists; thrombosis; coagulopathy; oral anticoagulant therapy; calcium metabolism; elastic fiber degradation; antioxidants; phenols

## 1. Introduction

Vitamin K is an essential fat-soluble vitamin that entails a class of structurally analogous compounds, which are defined in two main types: phylloquinone (K1 or PK, a plant-based form that exists primarily in leafy vegetables) and menaquinones (K2 or MK, a collection of isoprenologues mostly originating from bacterial synthesis, e.g., MK-7 or tissue-specific conversion of PK and/or menadione in animals and humans, e.g., MK-4) [1]. Vitamin K is a ligand for the nuclear steroid and xenobiotic receptor [2]. As illustrated in Figure 1, the vitamin K life cycle starts in the endoplasmic reticulum where dietary PK and MK forms of the vitamin undergo post-translational carboxylation of protein-bound glutamate (Glu) to γ-carboxyglutamate (Gla). Then, the vitamin K epoxide reductase enzyme reduces vitamin K into vitamin K hydroquinone 2 (KH2) [3]. Vitamin K acts as co-enzymes of γ-glutamyl carboxylase (GGCX), transforming under-carboxylated in carboxylated vitamin K-dependent proteins, e.g., carboxylated osteocalcin (cOC)/bone Gla protein (BGP) and matrix Gla protein (MGP) [4,5]. cOC promotes calcium transportation and fixation in bones. It also protects against age-related musculoskeletal deteriorations by promoting the release of adiponectin and the anti-inflammatory cytokine interleukin (IL)-10 and inhibiting inflammatory cytokines such as IL-6 and tumor necrosis factor-$\alpha$ [6]. cOC and cMGP protect against pulmonary and vascular elastic fiber degradation [4,7]. They also contribute to the regulation of glucose and lipid metabolism through the modulation of insulin signaling and adipokines. Additionally, they protect against vascular calcification through their ability to regulate vascular mineralization and prevent the formation of calcium crystals via a mechanism that involves inhibiting the transdifferentiation of vascular smooth muscle cells (VSMC), modulating calcification by VSMC-derived matrix vesicles and apoptotic bodies, and preventing calcium-phosphate precipitation [3,7–9]. The active form of vitamin K is involved in the coagulation cascade by regulating the synthesis of different clotting factors (II, VII, IX, and X) [10]. It also activates anticoagulant proteins C and S, which prevent the development of clotting by inhibiting the activity of essential coagulation factors [6].

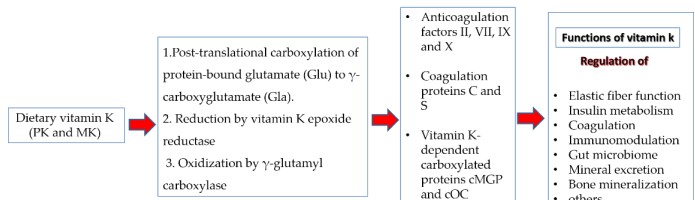

**Figure 1.** Schematic illustration of the vitamin K life cycle and common physiological functions of vitamin K. Dietary vitamin K is transformed to its active form following its post-translational carboxylation of protein-bound glutamate to γ-carboxyglutamate, reduction by vitamin K epoxide reductase, and oxidization by γ-glutamyl carboxylase. Active vitamin K contributes to vascular homeostasis by controlling the levels of the anticoagulant and procoagulant proteins as well as by fostering the carboxylation of various vitamin K-dependent proteins such as matrix Gla protein (MGP) and osteocalcin (OC). Carboxylated MGP and OC exert various bioactivities, e.g., regulating glucose and lipid metabolism, counteracting inflammation, protecting against vascular injury, etc.

Circulatory levels of PK depend on food intake, and they can be robustly measured, while circulatory MKs can be detected only when consumed in large quantities [5,11]. There is no single gold-standard biomarker of vitamin K status [5]. Circulatory levels of

undercarboxylated fractions of vitamin K-dependent proteins increase in response to vitamin K deficiency. Therefore, they signal body status regarding vitamin K. For instance, the inactive uncarboxylated, the dephosphorylated form of MGP, desphospho-uncarboxylated matrix Gla-protein (dp-ucMGP), acts as a key marker of vascular calcification associated with vitamin K deficiency in healthy individuals, diabetics, and renal/cardiac patients [12]. On the other hand, serum levels of dp-ucMGP [13] and plasma protein induced by vitamin K absence factor II (PIVKA-II), also known as des-γ-carboxy prothrombin [14], change in response to MK-7 supplementation and food intake, signifying their importance as markers of vitamin K status [13,15].

The recommended daily dietary allowance (60 microg/d) of vitamin K [16] is based on the hepatic requirement for clotting factor synthesis. However, it seems to be insufficient for an adequate vitamin K status since the levels of circulating uncarboxylated osteocalcin (ucOC) and dp-ucMGP rise in healthy individuals, especially in children and adults after the age of 40 [17]. Vitamin K contents in different foods vary considerably, which may affect the intake of such a vital vitamin. While most fruits are deficient in PK, some berries, green fruits, prunes [18], algae, and some vegetables (e.g., spinach, broccoli) are rich in PK [16]. Fermented soybean (natto) is a rich source of vitamin K [14]. In particular, its MK-7 content is quite high (939 microg/100 g) [16].

## 2. Possible Mechanisms for Increased Vulnerability to and Severity of COVID-19 in Vitamin K Deficient Groups

Coronavirus disease 2019 (COVID-19) patients display evidently low extrahepatic and hepatic levels of functional vitamin K, as reflected by higher levels of dp-ucMGP [4,19] and PIVKAII [9]. PIVKAII levels are higher among males, who represent the majority of COVID-19 patients than in females (72.3% vs. 36.8%) [9]. Vitamin K deficiency in COVID-19 is associated with the cytokine storm and severity of pulmonary insult [4,9]. In a cohort of 135 hospitalized COVID-19 patients, vitamin K deficiency was associated with a remarkably higher mortality compared with healthy controls (adjusted hazard ratio, AHR) = 1.50, 95% CI: 1.03; 2.18) [19]. In the same sample, vitamin D status did not vary between those with good and poor prognosis (defined as intubation and/or death), while accelerated elastic fiber degradation was higher in vitamin D sufficient persons than in those with mild deficiency (1,25-dihydroxy vitamin D; 1,25(OH)D > 50 nmol/L vs. 25–50 nmol/L) [20]. In another study, compared with healthy controls, COVID-19 patients showed similar 1,25(OH)D levels (25.8 vs. 21.9 ng/mL, $p = 0.09$) and higher dp-ucMGP levels (776.5 ng/mL vs. 549.8 ng/mL, $p < 0.0001$), along with increased odds of COVID-19 severity/mortality (adjusted odds ratio, AOR) = 1.84, 95% CI: 1.01; 3.45). Meanwhile, patients with vitamin D deficiency (<20 ng/mL) demonstrated a significant increase in dp-ucMGP levels (>780 ng/mL) and COVID-19 severity (AOR = 0.29 95% CI: 0.11; 0.67) [21]. Likewise, interesting results of a recent meta-analysis show that vitamin D deficiency is evident only in severe (not in mild) COVID-19 patients [22].

Vitamin D and vitamin K act synergistically to regulate the mineralization of blood vessels and the synthesis of bone matrix [23,24]. Both calcitriol and MKs increase the gene expression of BGP and MGP, BGP protein content in the extracellular matrix, while MKs enhance vitamin D3-induced bone mineralization. BGP binds hydroxyapatite to form a complex with collagen, while MGP promotes osteoclast differentiation and bone resorption [24]. However, more vitamin K is required for activating additional MGP and OC [23]. Given the critical interaction between vitamin D and vitamin K in calcium metabolism, along with their shared osteo-inductive activities and elastic fiber functioning, COVID-19 and its complications may be prevented or mitigated by a combined supplementation of vitamin D3 and vitamin K2 [20,23]. The next few paragraphs shed light on the involvement of vitamin K in increasing vulnerability to COVID-19 and COVID-19 pathology.

### 2.1. Lower Intake of Vitamin K Is Evident in Vulnerable Conditions

Severe COVID-19 commonly strikes old people and individuals with comorbidities such as diabetes, cardiovascular diseases, obesity, renal failure, etc. [25,26]. These

individuals are commonly deficient in vitamin K and other fat-soluble vitamins [7,27]. Epidemiological studies report large variations in the intake of foods containing vitamin K among certain age groups and population subgroups [7,16,27]. While the intake of food sources rich in vitamin K was high in 94% of young Japanese women [16], follow-up data from 3401 patients with chronic kidney disease show deficient intake of food containing vitamin K in 72% of the participants [27]. It seems that vitamin K plays a vital role in renal functioning—different levels of dietary vitamin K are associated with differences in the urinary excretion of various minerals. For example, the excretion of calcium and magnesium is lower in people consuming adequate levels of vitamin K and is higher in those with less consumption [14]. Given the role of vitamin K in renal functioning, its deficiency in COVID-19 might be associated with multiple organ failure, including liver and kidney injuries, which are common complications that increase fatality in SARS-CoV-2 victims [26,28,29].

### 2.2. Gut Microbiome Disintegration Depletes Gut-Derived MK in COVID-19 Vulnerable Groups

Gut microbiome dysregulation is one of the main causes of subclinical systemic inflammation in several chronic conditions, e.g., sarcopenia (age-related muscle loss/weakness), diabetes, obesity, malnutrition, depression, and other neurodegenerative diseases [30–33]. All these conditions experience deficiencies in microbiome-derived nutrients, and they are reported as high-risk factors for COVID-19 [28,30]. On the other hand, a considerable proportion of recovering COVID-19 patients develop conditions associated with alterations in the composition of resident gut microflora such as depression, muscle dystrophy, and cognitive impairment as long-term complications of this viral infection [31,34–36]. In this respect, prior alterations in the gut microbiome increase vulnerability to severe COVID-19 while the severity of COVID-19 contributes to the development of age-related disorders, which embroil gut-microbiome dysfunction as a key feature. Evidence confirms gut invasion by SARS-CoV-2 [37] and alteration in gut microbiome in COVID-19 victims. Such alterations persist even after viral clearance and symptom recovery [38,39]. Samples of the bronchoalveolar lavage fluid from patients with COVID-19 and community-acquired pneumonia are dominated by bacteria that commonly exist in the oral cavity and upper respiratory tract [40]. In fact, the gut-lung axis has also been implicated in cytokine production that underlies COVID-19 severity and pulmonary damage [40,41].

Vitamin K exists in a variety of food sources, signifying diet as a major source of vitamin K (predominately PK) [42]. Because of its multiple bioactivities, sufficient intake of vitamin K is considered a marker of a healthy diet and healthy gut [27]. In a sample of healthy young Japanese women, lower intake of vitamin k and iron was associated with a high relative abundance of Bacteroides and Clostridium and lower levels of health-promoting bacteria (e.g., Bifidobacterium) [43]. From another perspective, the human gut microbiome is considered the main source of MK [42]. Nonetheless, among various human microbial species that synthesize vitamins, microbial species involved in MK biosynthesis are reported to have the lowest abundance [44]. Because microbe-derived vitamin K is deficient in renal and age-related diseases (e.g., impaired bone strength and cognition) [42,45,46], it is reported that gut microbiome alteration in these conditions involves a reduction in MK-forming bacterial species in the gut, along with depleted levels of MK, especially longer chain isoforms, in the cecum, liver, kidney, and brain [42,45]. Thus, depleted production of gut-derived MK may be another reason for vitamin K deficiency in those groups when they contract COVID-19—research is needed to investigate such a possibility. On the other hand, intestinal invasion by SARS-CoV-2 may decrease the absorption of dietary vitamin K [47].

### 2.3. The Cytokine Storm, Hypoxia Associated with Pulmonary Damage, and the Interaction of Vitamin K and Its Proteins with SARS-CoV-2 Deplete Active Vitamin K

Investigations of host response biomarkers in plasma and bronchoalveolar samples obtained from mechanically ventilated COVID-19 patients demonstrate vivid systemic and, to a greater extent, pulmonary activation of inflammatory and procoagulant pathways.

The latter involves significant elevation in the levels of D-dimers, thrombin-antithrombin complexes, soluble tissue factor, C1-inhibitor antigen and activity levels, tissue-type plasminogen activator, plasminogen activator inhibitor type I, soluble CD40 ligand, and soluble P-selectin [48]. An autopsy examination reported presence of microthrombi along with foci of hemorrhage in the lungs and organs of a COVID-19 victim [49]. Vitamin K-dependent anticlotting proteins C and S play a major role in the control of microvascular coagulation and inflammation [6,50]. The combining of an endothelial surface transmembrane glycoprotein known as thrombomodulin with thrombin converts inactive protein C into its active form [50]. Activated protein C, along with protein S, modulates the inflammatory response, promotes endogenous fibrinolytic activity, and inhibits thrombin formation secondary to the inhibition of coagulation factors Va and VIIIa [50,51]. Sepsis in humans rapidly depletes protein C, resulting in activation of the procoagulant process, increased microvascular deposition of thrombin and fibrin, decreased blood supply and oxygen delivery to various organs, eventually leading to organ failure [50–52]. Inflammatory mediators potently inhibit vitamin K-dependent protein such as growth arrest-specific 6 (Gas6), resulting in augmentation of the inflammatory response [53]. Conditions associated with hypoxia, such as obstructive sleep apnea, exhibit excessive increases in plasma dp-ucMGP levels and bone turnover markers [54]. Inflammatory conditions, such as irritable bowel syndrome, are associated with elevated dp-ucMGP. On the other hand, dp-ucMGP exerts immunomodulatory properties and interferes with calcium metabolism, acting as a trigger of inflammation, vascular calcification, multiple physiological alterations associated with frailty, and mortality [55–57].

Although the exact mechanism of coagulopathy in COVID-19 is unclear, it encompasses an interplay of a network of related pathologies (Figure 2). Dysregulation of the coagulation cascade evoked by vitamin K deficiency is a documented major factor contributing to fatality in COVID-19. Consistent with other inflammatory conditions, vitamin K depletion in COVID-19 may be triggered by the cytokine storm and hypoxia associated with pulmonary damage [6]. In addition, inhibition of vitamin K epoxide reductase as a result of its binding to various non-structural proteins (NSPs) of SARS-CoV-2 [58,59] is likely to reduce vitamin K activation. This interaction may be more activated in individuals with polymorphism in the vitamin K epoxide reductase complex 1 (VKORC1) gene that is associated with a higher turnover of vitamin K [60]. Coagulants bind transmembrane protease serine 2 (TMPRSS2), the enzyme that primes the Spike (S) protein of SARS-CoV-2 as an initial step of viral invasion, more efficiently than Camostat mesylate [61]—a repurposed drug commonly used to treat chronic inflammatory disorders such as cancer and diabetes [62]. Thus, the binding of vitamin K itself to SARS-CoV-2 structural and non-structural proteins may deplete its levels [63–65]. As noted above, COVID-19 strikes vitamin K-deficient individuals. Gut invasion by SARS-CoV-2 may interfere with vitamin K absorption from the intestine [37,47]. It may also interfere with gut microbial structure resulting in diminution of nutrient-producing microbial species, which may lower the levels of gut-derived MK [38,39,42,45]. Deficiency of active vitamin K reduces cMGP and cOC, along with the accelerated accumulation of dp-ucMGP [10]. Thus, pneumonia-induced vitamin K depletion in COVID-19 entails vascular hyperpermeability due to severe endothelial insult, primarily of a prothrombotic nature [50–52]. Endothelial insult also involves direct invasion of the endothelium by SARS-CoV-2 via binding to its receptor, angiotensin-converting enzyme 2 (ACE2). This binding results in ACE2 downregulation and concomitant ACE 1 activation, leading to dysregulation of the renin-angiotensin system, which promotes the development of microcirculatory dysfunction [66,67]. Endothelial injury is furthered by cytokines and nitric oxides, which are excessively produced by activated macrophages secondary to activation of inducible nitric oxide synthase in severely ill patients [66]. Small vessel damage contributes to coagulopathy [66].

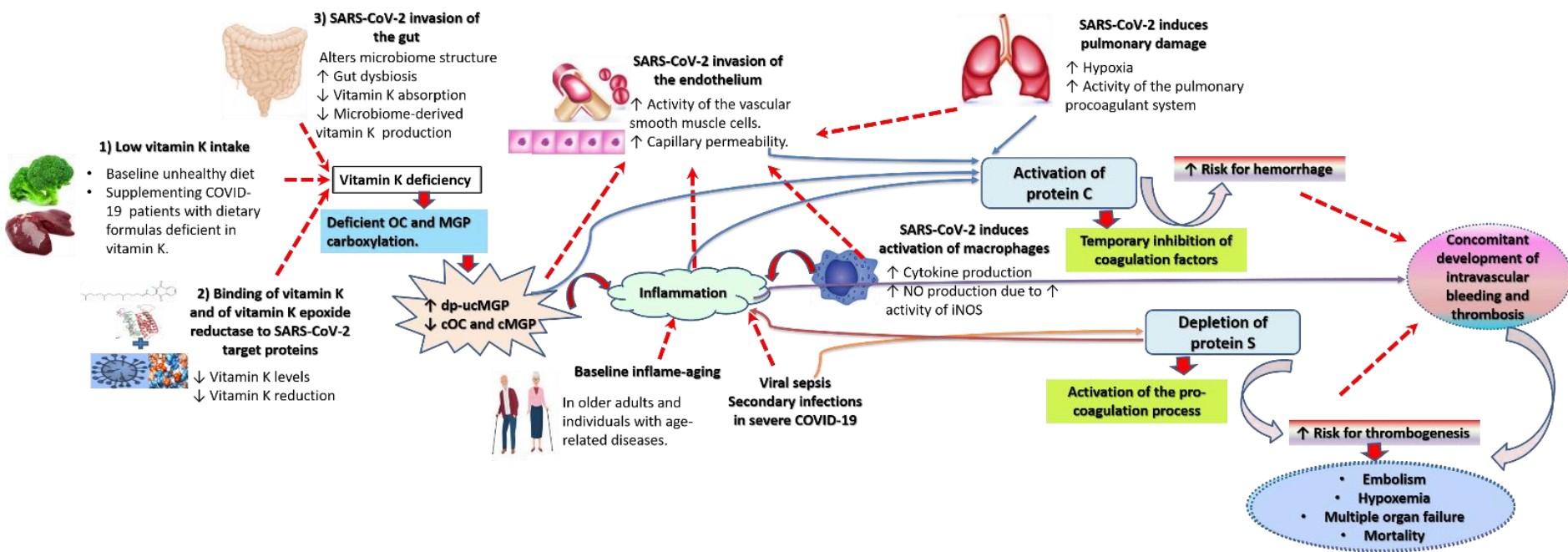

**Figure 2.** Schematic illustration of possible factors contributing to vitamin K deficiency in vulnerable groups and probable mechanisms of vitamin K deficiency in COVID-19-related thrombogenicity and its adverse effects. Vitamin K deficiency may occur due to (1) low dietary intake of vitamin K in COVID-19-prone individuals, and vitamin K-deficient formulas served to symptomatic COVID-19 patients; (2) the binding of vitamin K and vitamin K reductase to SARS-CoV-2 proteins; and (3) SARS-CoV-2 invasion of intestinal cells, resulting in microbiome alterations with reduced production of microbial vitamin K, gut injury resulting in accelerated systemic inflammation due to gut dysbiosis, and reduced absorption of dietary vitamin K. Deficiency of active vitamin K lowers the carboxylation of vitamin K dependent proteins such as osteocalcin (OC) and Matrix Gla Protein (MGP) resulting in increased levels of desphospho-uncarboxylated MGP (du-ucMGP). Upregulated du-ucMGP potentiates the inflammatory response and alters vascular mineralization. In addition, the binding of SARS-CoV-2 to its receptor in the endothelium induces direct vascular dysfunction. Meanwhile, hypoxia associated with pulmonary injury, as well as excessive levels of nitric oxide and cytokines released from activated macrophages, accelerate vascular injury. Microcirculatory disintegration triggers thrombin formation, resulting in activation of the anticoagulant protein C, which inhibits coagulation factors, resulting in an increased risk of bleeding. The cytokine storm and viral sepsis rapidly deplete the anticoagulant proteins; this is followed by subsequent activation of coagulation factors, e.g., Factor VIII. Accordingly, thrombogenesis within the context of microvascular bleeding occurs. Disseminated intravascular coagulation causes embolism and limits blood and oxygen supply to vital organs, resulting in multiple organ failure and fast progression to fatality in COVID-19 patients.

Activation of protein C occurs in response to vascular injury, increasing the tendency for bleeding [50–52]. The cytokine storm potentiates the prevalence of sepsis in COVID-19 patients, especially those who are old and nutritionally deficient [68]. Therefore, the levels of protein S may quickly drop due to SARS-CoV-2-induced septicemia, which is likely to activate procoagulants such as Factor VIII, fibrinogen, and D dimer [66,68]. Thus, the inflammatory milieu, along with dysregulation in the coagulation cascade, promotes a deadly positive thrombo-inflammatory feedback loop with a concomitant occurrence of thrombosis and hemorrhage in small vessels [6,69]. The pro-coagulant state in COVID-19, along with its related intravascular disseminated coagulation and embolization, evokes severe hypoxia, which causes the fast and fatal progression of the disease [66]. The mechanisms underlying alterations in proteins associated with vitamin K deficiency and their association with vascular injury and COVID-19 pathogenesis are described in detail in other more focused resources [4,8,9].

### 3. The Effect of Oral Anticoagulants, Including Vitamin K Antagonists (VKA), on the Prognosis of COVID-19

It is reported that during the course of COVID-19 infection, vitamin K antagonists (VKA)—inhibitors of vitamin K recycling such as warfarin (coumadin) and coumarins—affect procoagulant activity greater than the activation of extrahepatic vitamin K dependent proteins [70]. Anticoagulant agents may express a capacity to inhibit the cleavage of S protein and block viral endocytosis [61]. In an in silico investigation, various coumarins inhibited various NSPs of SARS-CoV-2 (MPro, NSP12, NSP15, NSP16) and interfered with S binding to ACE2 at an affinity greater than that of warfarin and some antiviral drugs such as favipiravir and hydroxychloroquine [71]. Therefore, VKA and other oral anticoagulants are expected to improve COVID-19 prognosis by reducing viral endocytosis and interrupting viral lifecycle [72].

The literature investigating the effects of VKA on the prognosis of COVID-19 in humans is mixed (Figure 3). A nation-wide study in Germany reports reduction in all-cause mortality and the use of invasive or non-invasive ventilation in hospitalized COVID-19 patients with pre-existing oral anticoagulation therapy including VKA (OR = 0.57, 95% CI: 0.40–0.83, $p = 0.003$) and direct oral anticoagulants (OAT; OR = 0.71, 95% CI: 0.56–0.91, $p = 0.007$)—but not with antiplatelet therapy alone (OR = 1.10, 95% CI: 0.88–1.23, $p = 0.66$) [72]. Another study recruited 427 patients, including 87 patients (19%) on OAT—54 patients (13%) received non-vitamin K oral anticoagulants (NOACs) and 33 (8%) received VKA. After adjustment for confounders, there were statistical differences in the development of acute respiratory distress syndrome and mortality between patients on anticoagulant therapy (either NOACs or VKA) and those not on anticoagulant therapy [73]. In a study involving 10 COVID-19 patients with a history of cardiac surgery, nine patients on chronic OAT recovered while one patient was on acetylsalicylic acid (ASA) and clopidogrel. While all patients had interstitial pneumonia (indicated by chest computed tomography), only the patient on ASA developed respiratory failure, was ICU-admitted for mechanical ventilation, and died because of cardiac arrest. The nine patients on chronic OAT recovered and were discharged [74]. In an investigation involving 2,377 Italian COVID-19 patients, including 125 patients on chronic OAT, the Charlson comorbidity index (CCI), evidently associated with mortality in COVID-19, was higher among OAT patients (4.35 ± 0.13) compared with non-OAT patients (3.04 ± 0.04). However, OAT-use was associated with less need for ICU admission [OR: 0.469 (0.250–0.880) vs. OR: 1.074 (1.017–1.134)] and less combined hard events [OR: 0.843 (0.541–1.312) vs. OR: 1.277 (1.215–1.342)]. On the contrary, OAT use was associated with higher mortality [OR: 1.756 (1.628–1.894)] vs [OR: 1.306 (0.78–2.188)] [75].

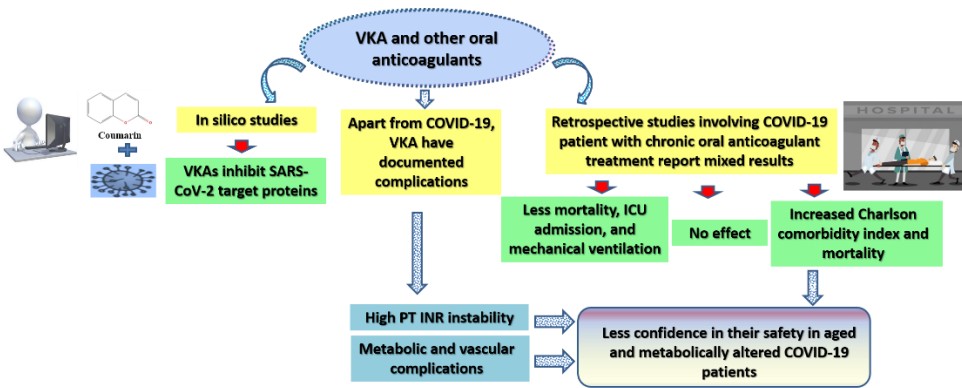

**Figure 3.** Summary of research findings on the possible effects of vitamin K antagonists (VKA) and other oral anticoagulants on COVID-19-related outcomes. While molecular simulation studies report efficient binding of VKA to target proteins of SARS-CoV-2, retrospective studies report inconsistent findings, with many of them pinpointing higher mortality in COVID-19 patients with a history of VKA treatment. In addition, the literature confirms the development of metabolic, vascular, and coagulopathy complications among non-COVID-19 patients treated with VKA. Thus, VKA may deteriorate the prognosis of COVID-19, given their vitamin K inhibitory activity.

Prior regular use of VKA in frail older adults hospitalized for COVID-19 was associated with increased short-term mortality (mortality at day 7 of hospitalization) [76]. Similarly, a retrospective investigation compared disease outcomes in 79 patients on OAT for any cardiac reason before the COVID-19 diagnosis with 1631 non-OAT-using patients. The need for intensive care, in-hospital stay, and mechanical ventilation, as well as mortality, were higher in the OAT group. Mortality was observed in 23 patients (29%) in the OAT group, and it was statistically higher than mortality in the non-OAT-using group ($p = 0.002$). The multivariate analysis noted that in-hospital mortality was associated with old age, male gender, lymphocyte count, procalcitonin, SaO2, and creatinine in OAT treatment [77]. In another study, analysis adjusted for age, gender, and comorbidity revealed no effect of OAT on mortality among 101 nursing home residents struck by COVID-19 [78].

In addition to the mixed reports on the role of VKA in COVID-19, antagonizing the effects of vitamin K by warfarin, a common VKA, is associated with increased risk for diabetes in patients with atrial fibrillation aged 65 years and above than non-vitamin K antagonist OAT, which act independently of vitamin K [79]. Low-risk atrial fibrillation patients using VKA show increased levels of coronary calcification [80]. The metabolic and vascular complications noted in these studies may justify the reported high mortality, probably due to sudden death, in COVID-19 patients receiving OAT despite their lower need for ICU admission [75]. Additionally, the use of VKA for four weeks in healthy volunteers was associated with a decrease in endogenous thrombin generation (ETP) and an increase in International Normalized Ratio (INR) as well as the concentrations of uncarboxylated factor II (ucFII), ucOC, and dp-ucMGP [13].

The mechanism of INR instability in VKA treatment involves VKA binding to vitamin K epoxide reductase enzyme resulting in inhibition of the recycling of inactive vitamin K epoxide back to its active reduced form. As a result, depletion of different coagulation factors develops within 72–96 h following VKA dosing, resulting in an anticoagulated state [10,81]. Among healthy volunteers receiving VKA, co-treatment with MK-7 (10, 20, and 45 μg day/6 weeks) significantly influenced anticoagulation sensitivity in some individuals, as noted by INR reduction in up to 60% of the participants, along with an increase in ETP by up to 30%. However, MK-7 had no effect on circulating ucOC and dp-ucMGP [13]. Systematic analyses involving observational studies and randomized trials suggest that the lower daily baseline intake of vitamin K increases the risk for INR instability in patients receiving VKA. Meanwhile, a high dietary intake of vitamin K

(>150 µg/day) may correct coagulation response in patients receiving VKA who exhibit INR instability. However, this effect was small and not clinically relevant [82,83].

It is important to note that some hospitalized COVID-19 patients on prolonged OAT and VKA demonstrate high instability of PT INR, which may be attributed to alterations in vitamin K metabolism (as illustrated above in Section 2.3), over/under treatment resulting from variations in fasting/dietary programs, co-medications (especially antibiotics), liver and cardiac injury, etc. [47,84]. Therefore, patients with hemorrhagic risk, especially those on antibiotics, who receive VKA need careful monitoring [47]. Because of lack of credible evidence on the effectiveness and complications of OAT, an undergoing RCT is evaluating if an adjunctive therapy with methylprednisolone and unfractionated heparin or with methylprednisolone and low molecular weight heparin (LMWH) is more effective in reducing any-cause mortality in critically-ill ventilated patients with pneumonia from SARS-CoV-2 infection compared with LMWH alone. No results are available yet [85].

## 4. Vitamin K Supplementation May Be Beneficial for Individuals Vulnerable to COVID-19

Several clinical trials involving healthy subjects from different age groups report positive effects of vitamin K supplementation on markers of vascular coagulopathy. In healthy adults aged 40–65 years, daily oral intake of MK-7 (180 µg and 360 µg/day for 12 weeks) dose-dependently reduced circulating dp-ucMGP by 31% and 46% three months after treatment compared with the placebo [15]. In a similar investigation, treating community-dwelling older adults with vitamin K1 (500 µg/day for 3 years) significantly decreased plasma dp-ucMGP levels. However, dp-ucMGP was not associated with vascular coagulation either at baseline or post-treatment [86]. Menopausal women receiving a multivitamin enhanced with 500 µg K1 for 3 years expressed a reduction in the progression of vascular coagulation compared with women receiving a multivitamin alone. Interestingly, serum MGP increased in K1-supplemented women and decreased in the control group. Nonetheless, it was not associated with the progression of vascular coagulation, which the authors justified by lack of quantification of the uncarboxylated and dephosphorylated forms of MGP in that study [87]. In a double-blind, placebo-controlled trial, 244 healthy menopausal women were randomized to receive MK-7 or a placebo for 3 years. MK-7 treatment decreased circulating dp-ucMGP by 50%, which was accompanied by a significant improvement in arterial stiffness [86]. An investigation involving supplementing healthy adults with K1, as well as low and high doses of MK-7 (<75 µg/day and >75 µg/day) for 12 weeks, reported a significant reduction in dp-ucMGP only in participants receiving the high MK-7 dose [88]. These findings suggest that MK-7 may protect against cardiovascular disorders by enhancing the carboxylation status of MGP [86,88]. Children (aged 10–12 years) and healthy adults (aged 40 years and above) who were supplemented with MK-7 (45 µg/day and 90 µg/day, respectively) responded better to treatment than their non-deficient counterparts; in terms of reduction of ucOC and dp-ucMGP. The latter decreased by 38% in both children and adults [17]. Altogether, these studies denote that higher oral doses of MK-7, rather than K1, can positively alter dp-ucMGP, in apparently healthy individuals, while even higher doses may be needed for those with deficiencies. A recently published trial investigating the effect of intravenous K1 in 52 critically ill patients with prolonged Owren PT report an increase in thrombin generation and the activity of coagulation factors II, VII, IX, and X, along with reductions in Owren PT, Quick PT, and dp-ucMGP at 20–28 h following treatment, albeit not to normal levels [89]. Thus, the route of administration of K1 and MK may affect their bioavailability and, subsequently, their therapeutic activities.

Several studies suggest that adequate intake of vitamin K-rich food may protect against age-related disorders and lower mortality associated with these conditions. Longitudinal data derived from 7216 Mediterranean participants show a significant decrease in the incidence of cancer, cardiovascular diseases, and all-cause mortality in individuals who increased their intake of vitamin K. These effects were most vivid in individuals prone to cardiovascular diseases [90]. Aggregate data from three cohorts involving 3891 participants (mean age 65 ± 11 y) uncovered an increased risk of all-cause mortality (not cardiovascular

disorders) in individuals with low circulating levels of PK [91]. Likewise, a long-term, large-scale follow-up study reported decreased all-cause mortality in patients with chronic kidney disease and cardiovascular diseases who consumed vitamin K at higher levels than recommended amounts [27]. In another longer-term cohort study following 33289 participants aged 20–70 years, the intake of large amounts of long-chain MKs was significantly associated with lower mortality in patients with coronary heart disease (CHD) [92]. In line, a meta-analysis including data from 222592 participants (in 21 longitudinal studies) reported a moderate reduction in the risk of CHD among individuals with higher dietary consumption of vitamin K [93]. Nonetheless, another meta-analysis cmprising 11 studies involving 33289 people with chronic disorders reported no association between either PK or MK with all-cause mortality and mortality from cardiovascular disease [94]. Based on the fact that cancer, CHD, and similar conditions predispose to COVID-19 and its high fatality, we may expect that vitamin K supplementation would lower the incidence of COVID-19 infection and fatality from COVID-19 among highly vulnerable groups [95].

## 5. Possible Potentials of Vitamin K as an Anti-COVID-19 Agent

Molecular simulation studies show that vitamin K2 can bind to the spike fatty acid site in SARS-CoV-2 and stabilize the closed conformation at an affinity higher than that of vitamin A and calcitriol, the bioactive form of vitamin D [65]. Vitamin K and its analog, Kappadione, inhibited NSPs necessary for the replication of SARS-CoV-2, such as the main protease (Mpro/3CLpro) and RNA-dependent RNA polymerase (RdRp) in silico [63,64]. Its affinity of binding (-32.8 kcal/mol) to PDB 6Y84 COVID-19 protease receptor is high, especially at S (Cys145) [96]. Computational studies indicate that the inhibition of Mpro/3CLpro by vitamin K results from its high stability in the pocket, which is associated with its ability to tightly bind the thiol group of Cys145 residue of Mpro/3CLpro [97]. In an in vitro study involving *Escherichia coli* cells infected with SARS-CoV-2, vitamin K3 expressed anti-COVID-19 activity by inhibiting Mpro/3CLpro in a dose-dependent manner—the best effect was obtained at a concentration of 7.96 μM [97]. Given that an intact activity of Mpro is necessary for viral replication [62], vitamin K supplementation may decrease viral load and associated tissue damage in vitamin K-deficient patients [28]. However, the binding of vitamin K to the target proteins of SARS-CoV-2 may cause extrahepatic insufficiency of vitamin K in COVID-19 patients [96]. Vitamin K deficiency is a major cause of dysregulation of extrahepatic proteins like endothelial anticoagulant protein S and related high thrombogenicity in severe COVID-19, even when the hepatic procoagulant activity is normal [70]. Thrombogenesis in COVID-19 is associated with disease severity, poor prognosis, and in-hospital mortality [47,59].

Vitamin K supplementation may protect against COVID-19. A machine-learning investigation of factors critical for COVID-19 occurrence, mortality, and case-fatality rates in all the states of the US and in 154 countries reports that higher consumption of vegetables, edible oils, proteins, vitamin D, and vitamin K lowers the risk for COVID-19 [95]. High-throughput yeast two-hybrid system and LUMIER assay showed that several coagulation related proteins, such as Poly(A) binding protein cytoplasmic 4 (PABPC4), serine/cysteine proteinase inhibitor clade G member 1 (SERPING1), and VKORC1, interact with NSPs2, NSPs 3, NSPs 13, NSPs 14, open reading frame 7 (ORF7), ORF3b, ORF14 of SARS-CoV [58]. Computational modeling confirms the interaction of PABPC4 and VKORC1 with ORF7 of SARS and SARS-CoV-2 [59]. In fact, carriers of a genetic polymorphism in VKORC1 (−1639A) exhibit resilience against COVID-19 as well as its thrombotic complications and severity—denoting that this allele may act as a low bioequivalent dose of VKA [70]. On the other hand, c.1173C > T polymorphisms, common in Africans and African Americans, are associated with warfarin resistance. Therefore, apart from the effects of age, diet, liver functioning, and general health, the interaction of coagulation proteins with SARS-CoV-2 can partially alter coagulopathy in certain COVID-19 patients [59].

## 6. Fermented Milk-Fortified with Bee Honey as a Proposed Source of Dietary MK and Other Bioactive Compounds for COVID-19-Prone and COVID-19-Struck Individuals

Given the uncertainty concerning the potential benefits of the anticoagulant medication against COVID-19 and related complications [75,77,78], the use of foods rich in vitamin K may be a safer approach to protect against coagulopathy among high-risk groups. Although most food composition databases are the most complete for PK, interest has been directed toward MKs because of their high bioactivities relative to PK. Nevertheless, food composition databases for MKs are not yet well-defined [5]. Herein, we summarize the literature for evidence describing MK in honey and fermented milk. Research on the protective effects of honey and fermented milk against the vascular and metabolic complications of VKA is totally lacking. However, within the light of the bioactivities of these products explored in this section, researchers are invited to investigate the possible protective effects of fermented milk or raw bee honey on the adverse effects of VKA. On the other hand, raw honey can mitigate SARS-CoV-2 activity in silico, in vitro, and in humans [62], while in silico studies report inhibitory activity of peptides derived from fermented milk against S protein [98]. In this section, we also explore the evidence suggesting that the sum of the bioactive properties of these two products in combination is greater than the bioactivities of honey and fermented milk on their own. Thus, we suggest that fortifying fermented milk with honey can be more effective in reducing vulnerability to COVID-19 than honey or milk alone.

### 6.1. Bee Honey as an Anti-COVID-19 Agent and a Natural Source of MK

Natural honey is a nutrient-rich bee product that expresses a wide range of pharmacological activities [33,99]. It has demonstrated inhibitory effects against SARS-CoV-2 in silico and in vitro, with evidence of its effectiveness recorded in hospitalized patients with moderate and severe COVID-19 infection [62]. However, only phenolic compounds in honey were examined in silico and in vitro, which implies that the anti-COVID-19 effects of honey are attributed to its phenolic compounds, with less information available on other bioactive compounds in honey such as proteins, vitamins (including MKs), minerals, and trace elements [62]. MKs have been recently detected in natural honey. Colloidal particles formed by active macromolecules in honey scatter the light and produce an elaborate UV spectral profile dominated by double absorption peaks at 240–250 nm, which signifies the stable honey conformation that promotes its internal production of hydrogen peroxide, which underlie the antibacterial activities of honey. MS chromatograms and UV spectral analysis of UPLC peaks identified compounds with UV λ (max) typical for naphthoquinones and mass ions differing by [M-68n]. The detected polyisoprene structure and the fragmentation patterns are identical to long-chain MKs, which were defined as a series of MK-3 to MK-7. MKs in honey are likely to contribute to its redox and antipathogenic activities [100]. A later investigation revealed that MKs in honey result from the shedding of the bacterial membrane of certain strains of lactic acid bacteria (LAB) resident in honey, such as *Bacillus subtilis* and *Bacillus cereus* [101].

### 6.2. Natural Milk as an Anti-COVID-19 Agent and a Natural Source of MK

Natural milk is rich in compounds capable of expressing an anti-SARS-CoV-2 activity. Treating goat milk whey fraction with trypsin produces beta-lactoglobulin-derived peptides (Ala-Leu-Pro-Met-His-Ile-Arg (ALMPHIR) and Ile-Pro-Ala-Val-Phe-Lys (IPAVFK)), which are reported to inhibit SARS-CoV-2 proteins in silico [102]. Another in silico investigation reported that peptides derived from whey and casein could potently bind human ACE2 and the receptor-binding domain of S protein, denoting peptides in fermented milk as potential inhibitors of SARS-CoV-2 invasion [98]. Because SARS-CoV-2 and bovine coronavirus (BCoV) are very close phylogenetically, researchers suggest that cross-immunity may develop through the consumption of cow milk immune to BCoV [103]. BCoV cow milk may act as a particular natural vaccine where anti-BCoV antibodies present in milk may facilitate antigenic recognition of some highly conserved structures of viral proteins, particularly

M and S2, resulting in a total or partial inhibition of SARS-CoV-2 [103]. Others suggest that drinking microfiltered raw immune milk or colostrum from cows vaccinated against SARS-CoV-2 may facilitate immunoglobulin transfer and passive immunity—protecting humans against SARS-CoV-2 [104]. This notion may be scientifically sound given that research has identified SARS-CoV-2 antibodies in the breast milk of women with confirmed COVID-19 [105]. Most of these antibodies (88%) are spike-specific of high titer (50%), and in vitro, they expressed neutralizing effects against spike-pseudotyped VSV (IC50 range, 2.39–89.4 ug/mL) [106]. SARS-CoV-2-specific antibodies can be detected in breast milk for up to 10 months following the first appearance of COVID-19 symptoms [106,107]. W-containing peptides, which are rich in tryptophan, are common in milk fermented with certain LAB species. They are reported to inhibit ACE2 [33], which is essential for viral endocytosis in COVID-19.

A previous study has identified vitamin K in modest amounts in whole milk and trivial amounts in skim milk. Therefore, different milk formulations are frequently fortified with synthetic vitamin K [108]. *Lactobacillus fermentum LC272* isolated from raw milk is highly resistant to gastric and bile juices, antibiotics, and pathogenic bacterial species. Their use as a starter culture is associated with a considerable increase in the milk content of vitamin K [109]. Processed cheeses, not involving fermentation in their preparation, as well as fat-free dairy products have low vitamin K content. On the contrary, MK9, MK10, and MK11 are abundant in full-fat dairy products, especially in fermented cheeses. Meanwhile, PK, MK4, MK8, and MK12 are detected in modest amounts [110]. Milk fermented with kefir contains MK-9 (5/µg/100 g) [111]. Milk and whey are reported to promote the growth of LAB, which produce MK [112,113]. These reports signify a key role of fermentation in increasing the content of MK in milk, which makes fermented dairy products a common source of vitamin K2 [1].

An investigation of MK in various fermented dairy products reported a wide diversity of MK contents ranging from undetectable to 1100 ng/g of product. MK also forms a considerable variety, with MK-9 exhibiting the highest occurrence—around four times greater than MK-8, the second most abundant MK. The abundance of MK-9 is highly associated with that of MK-8, pinpointing that they are produced by the same bacterial species. Such variations in MK content and forms are highly dependent on the bacterial species and differences in milk fermentation technologies [114]. Genetic modifications involving simultaneous overexpression of mvk, preA, and menA genes in *Lactococcus lactis*, the main source of MK in the Western diet, resulted in a considerable increase in MK production. The use of MK-overproducing strains for milk fermentation effectively increased milk content of MK by 3-fold compared with the wild type [115]. Children receiving a fermented milk beverage-fortified with iron amino acid chelate (3 mg iron per 80 mL) and *Lactobacillus acidophilus* expressed significant improvements in hematological parameters and nutritional status compared with children receiving unfortified fermented milk as a placebo. Serum ferritin increased in the placebo group [116]. This finding emphasizes the contribution of MK-producing bacterial species to the therapeutic activities of fermented milk.

### 6.3. Pharmacological Properties of Fermented Milk Fortified with Honey and Their Relevance to COVID-19

The term "milk and honey" has been used in the sacred script to refer to ecstasy, wealth, and endearment [117]. In traditional Chinese medicine, milk and honey are used in Paozhi processing—treating raw medicinal herbs that are cut into decoction pieces with honey or milk—as extraction solvents to increase their yield of bioactive substances [118]. This may explain why most ancient Egyptian medicines frequently comprised a combination of honey, milk, and wine [119]. Recent research confirms that bee honey and milk in combination represent a source of high nutritional value and antimicrobials, which grant protection against a plethora of pathogenic infections [120]. This section explores the potential bioactivities of fermented milk with honey supplementation. Accordingly, we

speculate that fortifying fermented milk with honey may be beneficial for protecting against COVID-19 through various ways, including enhancing MK yeild (Figure 4).

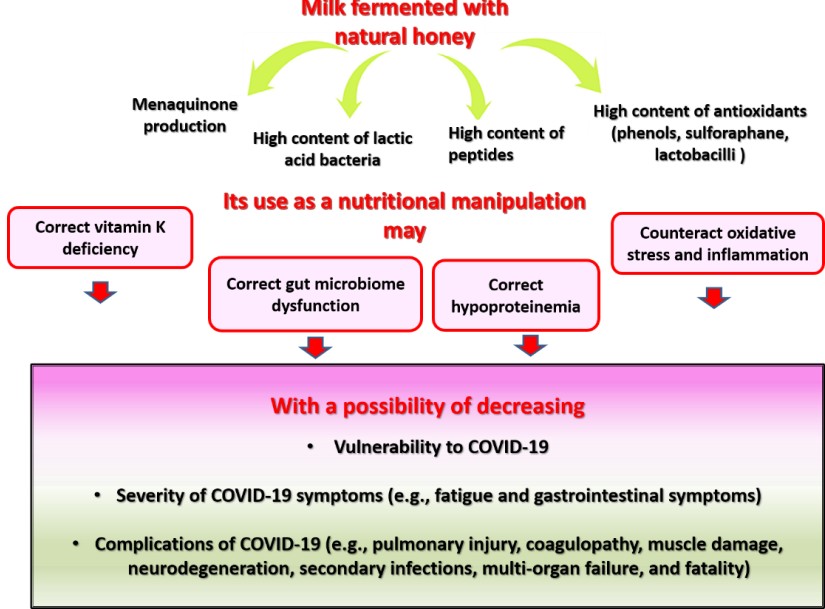

**Figure 4.** Examples of beneficial compounds in fermented milk fortified with honey and their potential benefits in COVID-19. Fermented milk supplemented with natural honey is probably rich in bacterial-derived vitamin K, lactobacilli, and their metabolites, peptides, and antioxidant compounds such as phenols. Its use as a dietary manipulation in immune-compromised individuals (elders and those with co-morbidities) may enhance their immune response, granting them more resistance to SARS-CoV-2. Its use as a dietary supplement in non-severe COVID-19 patients may promote a favorable disease prognosis.

### 6.3.1. Fermented Milk Supplemented with Honey as a Source of Probiotics and Bacterial MK

Food fermentations that synthesize MK can be initiated by numerous bacterial species used in starter cultures. Research has recently focused on enhancing MK production in dairy fermentation through the identification and selection of MK-producing bacteria [1]. Nowadays, yogurt enriched with bioactive compounds such as bee honey is available in most local markets in many parts of the world. This yogurt is intended to receive a better consumer acknowledgment [121]. This is because fortifying fermented milk with honey is likely to enhance its bioactive properties, mainly by enhancing its probiotic content. Honey is rich in different strains of probiotic LAB, making it an appealing starter culture [101,113]. Oligosaccharides, which comprise 5–10% of honey content, are non-digestible prebiotics that inhibits the growth of pathogens and provide favorable conditions for the growth of LAB and improve their fermentation metabolite profile [62,122]. Therefore, yogurt containing honey is likely to supply the intestinal microbiome with prebiotic oligosaccharides to boost their metabolic activities [122]. Using milk as a medium for fermentation would increase the survival of LAB because milk can protect LAB against oxidative stress associated with reactive oxygen species, which is produced by these bacteria [113]. Numerous studies show that honey can accelerate milk fermentation and improve LAB growth over a long period (21 days of storage at 4 °C) [112,122], without disrupting the survival of probiotic species such as *Lactobacillus reuteri* [112,122,123].

LAB contributes to MK biosynthesis through the activity of several enzymes encoded by *men* genes in the shikimate pathway resulting in the generation of a naphthoquinone ring from chorismate. Demethylmenaquinone (DMK) forms when the isoprenoid side chain, which is synthesized separately, joins the naphthoquinone ring. MK synthesis

results from DMK methylation [1]. Thus, honey-fortified yogurt represents a functional probiotic food [123], which might ameliorate vitamin K deficiency because of its MK content. Its probiotic content may have implications for the prevention and treatment of COVID-19. A meta-analysis involving an association network concerning 90 identified training genes linked with COVID-19 and probiotic treatment reported that probiotics can therapeutically interfere with ACE2-mediated viral invasion, activation of the systemic immune response, Nlrp3-mediated immunomodulatory pathways, immune cell migration that cause pulmonary damage and cardiovascular difficulties, and altered glucose/lipid metabolic pathways in the disease prognosis [40].

### 6.3.2. Fermented Milk Supplemented with Honey as a Source of Antioxidants

Efficient fermentation increases the antioxidant content in foods [124,125]. In accordance, honey-fortified yogurt expresses potent antioxidant properties [121,126], probably due to enhanced milk fermentation and probiotic content (Section 6.3.1). Research denotes that the level of the antioxidant activity displayed by yogurt fortified with honey depends on yogurt levels of casein haplotypes ($\alpha$s1-, $\beta$-, $\kappa$-casein)—which are related to its LAB activity [126]. W-containing peptides in fermented milk are involved in the synthesis of nicotinamide adenine dinucleotide (NAD) and phosphorylated NAD (NADP), which are key effectors in redox regulation [33].

Unfortified yogurt contains some phenols derived from feed [121]. However, polyphenols and flavonoids in bee honey are likely to augment the antioxidant bioactivity of milk [121,122,126]. Enriching yogurt with herbal honey such as *Nigella sativa* L. is reported to cause an exponential increase in its total phenolic compounds—1415.0 mg GAE/kg compared with 202.5 mg GAE/kg in unfortified yogurt [121]. Accordingly, significant differences in the antioxidant activities of fortified and unfortified yogurts were recorded on day one and day 28 of refrigerated storage [121]. Nonetheless, different types of honey vary in their phenolic content—the ability of honey, with higher phenolic content, to increase the antioxidant activity of yogurt is significantly greater than honey low in phenols [122,126].

Natural honey from beehives contains sulforaphane, a dietary isothiocyanate, with potent antioxidant activity [127]. Feeding bee-workers sugar extract containing sulforaphane extracted from sulforaphane-rich plants, such as broccoli, may considerably increase the honey content of sulforaphane [128]. Treating human colon cancer HCT116 and SW480 cells with a combination of sulforaphane and Lactobacilli, such as those found in fermented milk, are reported to accelerate apoptotic reactions [129]. In human Caco-2 cells, sulforaphane treatment increased the activity of nuclear factor (erythroid-derived 2)-like 2 (Nrf2) by promoting demethylation of the Nrf2 promoter region, providing chemoprevention against colon cancer [130]. Nrf2 is the master antioxidant signaling pathway in humans [32] and can inhibit the AT1 R axis [124]. Lactobacilli in honey and fermented foods are also potent Nrf2 activators [124,129]. Therefore, it might be expected that treating fermented milk with honey would heighten its antioxidant potential due to the synergetic interaction between lactobacilli (in milk and honey) and sulforaphane available in honey.

The antioxidant properties of fermented milk with honey can be of particular importance within the context of COVID-19. The binding of SARS-CoV-2 to ACE2 downregulates ACE2 resulting in potentiation of the angiotensin II receptor type 1 (AT1 R) axis, which accelerates oxidative stress, insulin resistance, pulmonary and endothelial damage in COVID-19 [124]. In addition, oxidative stress is high in COVID-19 patients with cytokine storms secondary to abnormal activation of several types of immune cells [62]. Fermented food is high in antioxidants, and its consumption is associated with low COVID-19-related death rates [124,125]. Therefore, we speculate that the fermented milk-honey mixture may exert similar effects.

### 6.3.3. Fermented Milk Supplemented with Honey as a Source of Antimicrobial Agents

Experimentally, fortifying yogurt with sulforaphane extracted from broccoli sprouts, which also exists in natural honey, has no inhibitory effect on beneficial gut microbial

species such as *Bifidobacterium lactis* and *Lactobacillus acidophilus*. In the meantime, they can exert a synergistic inhibitory effect on the growth of Helicobacter pylori [131]. Honey is capable of correcting gut-microbiome dysfunction and related dysbiosis because of the antipathogenic activities exerted by its hydrogen peroxide content, oligosaccharides, and polyphenols [132,133]. Therefore, the intake of honey-milk mixture may correct gut microbial alterations and related gastrointestinal symptoms, which act as the main source of malnutrition in COVID-19 patients [25,28].

Using a mixture of honey and yogurt is reported to mitigate vaginal fungal infection [134]. The antifungal effect of these products may have implications for the prevention of secondary fungal infections in COVID-19. This hypothesis is based on evidence showing that the propagation of opportunistic fungal species (e.g., *Candida albicans*, *Candida auris*, *Aspergillus flavus*, and *Aspergillus niger*) is common in the gut of COVID-19 patients, and it is associated with leaky gut [39]. Likewise, COVID-19 patients on mechanical ventilation display an increased incidence of pulmonary fungal infection with *Aspergillus fumigatus*, which is associated with a considerable increase in mortality [135].

### 6.3.4. Fermented Milk Supplemented with Honey as a Source of Bioactive Peptides and Proteins

In addition to being a source of vitamin K, milk supplemented with honey (4%) when fermented by *Lactobacillus helveticus MTCC5463* demonstrates a maximum production of bioactive peptides [136]. As noted above, peptides in fermented milk promote signaling involved in redox balance [33]. Thus, the fermented honey-milk mixture may represent a sound source of protein for symptomatic COVID-19. These patients usually undergo excessive protein loss due to SARS-CoV-2-induced inflammation/oxidative stress, metabolic imbalance, as well as reduced food intake and nutrient loss due to gastrointestinal injury [25,28,31].

Protein degradation in severe COVID-19 infections contributes to inflammation, which affects different body structures, including skeletal muscle proteins leading to weakness, muscle aches, and fatigue. The latter is the third most common symptom in symptomatic patients [25,26,31,137]. In severe cases, hypoproteinemia boosts multiple organ failure and mortality [25,26,28]. On the other hand, high protein/amino acid supplementation to COVID-19 patients may enhance immunity, decrease hospital stay, prevent muscle damage, reduce fatigue, and lower mortality [25,28].

### 6.3.5. Anti-Fatigue and Anti-Sarcopenic/Anti-Cachectic Properties of Fermented Milk Supplemented with Honey

The nutritive, antioxidant, and anti-inflammatory properties of bee honey are reported to support its potential as a natural anti-fatigue agent [138]. Similarly, fermented milk exerts anti-fatigue effects in humans [139–141]. The anti-fatigue properties of fermented milk are largely related to its LAB content [140,141], which involve increased milk protein content and antioxidant properties. Moreover, experimental studies associate the anti-fatigue effect of fermented milk with inhibition of the growth of endotoxic bacteria as noted by reduction of *Firmicutes/Bacteroidetes* ratio in mice treated with kefir (milk fermented with special grains and yeast), which was associated with an enhanced capacity for exercising [142]. Therefore, supplementing COVID-19 patients with a combination of fermented milk and honey may be expected to decrease fatigability due to the anti-fatigue properties of these natural products. Moreover, this combination may also limit muscle dystrophy (a key cause of fatigue in moderate to severe COVID-19), given that both honey and other bee products are reported to exert anti-cachectic and anti-sarcopenic properties [132,133].

Proteins in milk and whey may also counteract pathologies that underlie skeletal muscle damage, resulting in the restoration of muscle mass and muscle strength [31,143]. Metabolic disorders are associated with low peripheral insulin sensitivity and muscle wasting. These conditions increase vulnerability to COVID-19 and its related cachexia. Meanwhile, SARS-CoV-2 invasion activates multiple signaling cascades that alter metabolism in order to create conditions favorable for viral replication [28,31,62]. Both honey and

fermented milk are capable of correcting metabolic dysfunctions [33,144]. Reduction of *Firmicutes/Bacteroidetes* ratio by probiotics (such as those found in honey and fermented milk) is associated with improvement in metabolism and reduction in toxic metabolites such as chenodeoxycholic acid, deoxycholic acid, lithocholic acid, d-talose, and N-acetyl-glucosamine, along with decreased depletion of d-glucose and l-methionine, as well as upregulation of the metabolism of pyruvate, retinol, and PPAR in the liver [145].

### 6.3.6. Neuroprotective Properties of Fermented Milk Supplemented with Honey

Experimentally, natural honey is reported to protect astrocytes against oxidative stress, which may be associated with neuroprotection against neurodegeneration [99]. Cumulative knowledge shows that bee honey exerts antidepressant, anxiolytic, and cognitive-enhancing properties in experimental animals and humans [138,144,146,147]. Fermented milk is reported to improve cognitive functioning in experimental animals with induced cognitive dysfunction [148], Alzheimer's disease transgenic mice [149], and healthy older adults undergoing cognitive fatigue [141]. Its cognitive promoting activities involve modification of gut microbial composition, reduction of beta-amyloid deposition in the cortex and the hippocampus, increasing the expression of brain-derived neurotrophic factor, and inhibiting hippocampal p-extracellular signal-regulated kinase (ERK)/ERK, p-cyclic adenosine monophosphate (cAMP)-response element (CRE)-binding protein (CREB)/CREB [149]. Dysregulation of hippocampal ERK/CREB signaling is associated with the increased stress of the endoplasmic reticulum, activation of inflammatory signaling, dysfunctional autophagy, and altered production of neurotrophins, which all contribute to neurodegenerative disorders [32,150]. Enriching fermented milk with resveratrol and organic selenium is reported to contribute to its cognitive-promoting properties [149]. Evidence associates COVID-19 acquisition in mentally-intact individuals with the development of neurodegenerative disorders (e.g., depression, cognitive impairment, etc.) post-recovery. The occurrence of these disorders is higher in older patients [35,36,151,152]. In this respect, older and severe COVID-19 patients may benefit from the neuroprotective effects of honey and fermented milk, albeit research is needed to support this hypothesis.

### 7. Conclusions

The binding of vitamin K and its related proteins to various target proteins of SARS-CoV-2 in silico and in vitro highlights its potential as an inhibitor of SARS-CoV-2. In line, humans with a high intake of vitamin K have less predisposition to COVID-19. Because vitamin K regulates the immune response, coagulation, and elastic fiber function, its deficiency is associated with cytokine surge, vascular injury, dysregulation of the coagulation cascade, and thrombo-inflammatory complications that contribute to high fatality in vulnerable individuals struck by COVID-19 (older adults and chronic disorders). Likewise, VKA treatment may interrupt vitamin K life cycle and interfere with the function of vitamin K-dependent proteins, rendering the small vessels prone to injury. Therefore, bleeding tendency and fatality are high in hospitalized COVID-19 patients on chronic VKA treatment. Bee honey is rich in various health-promoting elements, including probiotic bacteria that can produce MK. The literature documents inhibitory effects of bee honey against SARS-CoV-2 in silico and in vitro, along with improved treatment outcomes in hospitalized COVID-19 patients. Fermented milk is rich in numerous bioactive compounds, including MKs. It has a plethora of bioactivities, albeit not empirically tested within the context of COVID-19 except in few in silico investigations. Fortifying fermented milk with natural honey may increase milk contents of the bioactive form of vitamin K, bioactive peptides, antioxidants, probiotics, and antimicrobial agents (e.g., oligosaccharides). All these compounds have the potential to enhance immunity and correct nutritional deficiencies, which are commonly expressed in COVID-19 infection. Accordingly, based on these premises in Section 6.3, we speculated that the use of fermented milk-fortified with honey as a dietary supplement may protect against COVID-19 and mitigate the severity of the disease. Experimental investigations are warranted to test this proposed hypothesis.

**Author Contributions:** Conceptualization, A.M.A., H.K. and A.O.H.; software, S.A., M.E.A. and A.O.H.; validation, A.M.A. and A.O.H.; resources, A.M.A., H.A.A., S.A. and A.S.M.; writing—original draft preparation, A.M.A., A.O.H., H.A.A., A.S.M., M.E.A. and S.A.; writing—review and editing, A.M.A. and H.K.; visualization, A.M.A. and A.O.H.; supervision, A.M.A. and H.K.; project administration, A.O.H. All authors have read and agreed to the published version of the manuscript.

**Funding:** This research received no external funding.

**Conflicts of Interest:** The authors declare no conflict of interest.

**Abbreviations**

| | |
|---|---|
| 1:25(OH)D | 1,25-dihydroxyvitamin D |
| AT1 R | Angiotensin II receptor type 1 |
| AHR | Adjusted hazard risk |
| CHD | Coronary heart disease |
| DMK | Demethylmenaquinone |
| dp-ucMGP | Dephosphorylated-uncarboxylated matrix Gla protein |
| ETP | Endogenous thrombin |
| GGCX | γ-glutamyl carboxylase |
| INR | International Normalized Ratio |
| LAB | Lactic acid bacteria |
| LMWH | Low molecular weight heparin |
| MGP | Matrix Gla protein |
| MK | Menaquinones |
| Mpro | Main protease |
| NAD | Nicotinamide adenine dinucleotide |
| NADP | Phosphorylated NAD |
| NOACs | Non-vitamin K oral anticoagulants |
| Nrf2 | Nuclear factor (erythroid-derived 2)-like 2 |
| NSPs | Non-structural proteins |
| OAT | Oral anticoagulants |
| ORF7 | Open reading frame 7 |
| PABPC4 | Poly(A) binding protein cytoplasmic 4 |
| PIVKAII | Prothrombin induced by vitamin K absence-II |
| PK | Phylloquinone |
| RdRp | RNA-dependent RNA polymerase |
| SERPING1 | Serine/cysteine proteinase inhibitor clade G member 1 |
| S protein | Spike protein |
| TMPRSS2 | Transmembrane protease serine 2 |
| VKA | Vitamin K antagonists |
| VKORC1 | Vitamin K epoxide reductase complex 1 |
| ucFII | Uncarboxylated factor II |
| ucOC | Uncarboxylated osteocalcin |

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
