# Peer review of "Vitamin K in COVID-19—Potential Anti-COVID-19 Properties of Fermented Milk Fortified with Bee Honey as a Natural Source of Vitamin K and Probiotics"

_fermentation, doi:10.3390/fermentation7040202_

Round 1
Reviewer 1 Report
Ali et al. start by describing in their review article, the current evidence that is available about the potential role of vitamin K and vitamin K antagonists in COVID-19. Then the article type switches from being a review to being a perspective article. In the latter part of article, they speculate about a potential role of bee honey in the prevention of (severe) COVID-19. This part of the article can be regarded as a Perspective rather than a Review.
I found their speculations on milk fortified with bee honey in COVID-19 - in relation to vitamin K - rather random. However, it was surprising and interesting to read about vitamin K in COVID-19 from a different perspective.
Minor comments:
- With the phrase in the abstract "even in patients with vitamin D sufficiency", the authors suggests that a role of vitamin D in the pathogenesis of COVID-19 is established, which is absolutely not the case. This phrase should be deleted.
- On page 6 of 18 section 3, the authors discuss the fact that INR has often remarkable instability in COVID-19 patients. They state that this may be due to alterations in vitamin K metabolism (which is in my opinion the most likely cause). It would be interesting to further speculate on the mechanisms responsible for this.
Major comments:
- I don't like the title. It suggests that it has been postulated by others that vitamin K antagonists have anti-COVID effects. In contrast, it has been postulated that vitamin K itself would protect against severe COVID-19.
- On page 2 (sentence 67-68), the authors write "vascular calcification through the modulation of NF-KB". That is not how MGP and OCN protect against calcification. I suggest that the authors read a review about MGP (such as https://pubmed.ncbi.nlm.nih.gov/23375872/) and rewrite this section.
- On page 3 of 18 (paragraph 2), the authors discuss dp-ucMGP and PIVKA-II in COVID-19. What do the authors mean with "peripheral tissue"? Dp-ucMGP is regarded as a marker of extrahepatic vitamin K status and PIVKA-II as a marker of hepatic vitamin K status. This section should be rewritten.
- On page 3 of 18 (paragraph 2), the authors state that "1,25(OH)D is antagonized by vitamin K. This is not correct. What happens is that 1,25(OH)D upregulates vitamin K synthesis (probably as a natural protective mechanism against the calcifying effects of 1,25(OH)D) and subsequently more vitamin K is needed to activate the additional MGP.
- On page 4 of 18 sentence 155, the authors write "MK is reported... ...biosynthesis pathway present." I do not understand this sentence. It should be rewritten.
- On page 6 of 19 sentence 266-269, the authors write "Children (10-12 years)... ... in both groups". I have no clue what the authors mean with those sentences.
Author Response
Manuscript ID: fermentation-1309224
Response to the comments of Reviewer 1
We are very much grateful for the time, sincere help, and patience of the Reviewer. We have modified the whole manuscript taking into account the comments of the Reviewer, which are addressed line-by-line as shown below. Replies come underneath in red.
I found their speculations on milk fortified with bee honey in COVID-19 - in relation to vitamin K - rather random. However, it was surprising and interesting to read about vitamin K in COVID-19 from a different perspective.
We agree with the reviewer; reports on vitamin K in fermented milk were limited. In this version, we included more reports on vitamin K in milk, stressing the role of fermentation in increasing milk content of vitamin K (lines 532-560).
Although limited evidence supports the anti-COVID-19 potential of bee honey, no studies have reported on the protective effects of fermented milk against COVID-19. Therefore, we noted in different instances in the text that we “speculate” that fermented milk and honey in combination would increase exert anti-COVID-19 benefits.
In the current version, we attempted to organize this section by adding subheading that address common themes that address the bioactivities of fermented milk with honey, along with their possible relevance to COVID-19.
Minor comments:
- With the phrase in the abstract "even in patients with vitamin D sufficiency", the authors suggests that a role of vitamin D in the pathogenesis of COVID-19 is established, which is absolutely not the case. This phrase should be deleted.
We agree with the reviewer. Accordingly, we have deleted that phrase.
- On page 6 of 18 section 3, the authors discuss the fact that INR has often remarkable instability in COVID-19 patients. They state that this may be due to alterations in vitamin K metabolism (which is in my opinion the most likely cause). It would be interesting to further speculate on the mechanisms responsible for this.
We elaborated on the mechanism of INR instability in VKA treatment (line 356-360) and attempted to explain the dynamics of alterations in vitamin K metabolism in COVID-19 infection (line 198-286).
Major comments:
- I don't like the title. It suggests that it has been postulated by others that vitamin K antagonists have anti-COVID effects. In contrast, it has been postulated that vitamin K itself would protect against severe COVID-19.
Yes, we have modified the title: 1) removing the phrase suggesting that vitamin K antagonists have anti-COVID effects and 2) highlighting the potential anti-COVID-19 properties rather than effects of fermented milk because of scarcity of experimental studies.
- On page 2 (sentence 67-68), the authors write "vascular calcification through the modulation of NF-KB". That is not how MGP and OCN protect against calcification. I suggest that the authors read a review about MGP (such as https://pubmed.ncbi.nlm.nih.gov/23375872/) and rewrite this section.
On page 3 of 18 (paragraph 2), the authors discuss dp-ucMGP and PIVKA-II in COVID-19. What do the authors mean with "peripheral tissue"? Dp-ucMGP is regarded as a marker of extrahepatic vitamin K status and PIVKA-II as a marker of hepatic vitamin K status. This section should be rewritten.
Yes, thank you very much. We have rewritten this section (line 76-79).
- On page 3 of 18 (paragraph 2), the authors state that "1,25(OH)D is antagonized by vitamin K. This is not correct. What happens is that 1,25(OH)D upregulates vitamin K synthesis (probably as a natural protective mechanism against the calcifying effects of 1,25(OH)D) and subsequently more vitamin K is needed to activate the additional MGP.
Yes, thank you very much. We have rewritten this section (line 135-142).
- On page 4 of 18 sentence 155, the authors write "MK is reported... ...biosynthesis pathway present." I do not understand this sentence. It should be rewritten.
Yes, we have rewritten the statement to make it clearer (line 186-187).
- On page 6 of 19 sentence 266-269, the authors write "Children (10-12 years)... ... in both groups". I have no clue what the authors mean with those sentences.
Yes, we have rewritten the statement to make it clearer (line 406).
We hope that the manuscript has been satisfactorily modified and that the current version will be suitable for publication.
Best regards,
Corresponding author

Reviewer 2 Report
Review of the Article:
Vitamin K in COVID-19—potential anti-COVID-19 effects of vitamin K
antagonists (VKA) and fermented milk fortified with bee honey as a natural
source of vitamin K and probiotics.
In this article, the authors investigated the potential of vitamin K antagonists (VKA) and fermented milk fortified with bee honey as a natural source of vitamin K and probiotics. Supplementing fermented milk with honey increases milk peptides, bacterial vitamin K production, and compounds that act as potent antioxidants: phenols, sulforaphane, and metabolites of lactobacilli. They conclude that fermented milk that contains natural honey can be a dietary manipulation capable of correcting nutritional and immune deficiencies that predispose to and aggravate COVID-19.
Important remarks:
I strongly recommend for this article be revised by a native English speaker since some paragraphs are badly written.
Minor remarks:
Regarding this paragraph “To support this view, we synthesized the literature for the association between vitamin K and SARS-CoV-2 based on the findings reported from machine learning studies, molecular simulation, and human studies.”
What do you mean by synthesizing the literature?
Figure 1:
In “Figure 1. Schematic illustration of common physiological functions of vitamin K, possible mechanisms of its deficiency in COVID-19, and associated adverse effects in deficient patients.”
This description is wrong since no mechanisms are shown to demonstrate deficiency. The figure is very uninformative and the colors are not used to guide the read and interpretation of the figure but are randomly chosen.
Figure 2:
In this case, the figure is difficult to follow it has 3 arrows misplaced and the function of the blue box surrounding an area is incomprehensible.
Figure 3:
The colors make the figure difficult for the reader, they are distractive because the reader is trying to understand if the colors have a meaning in describing the figure.
Conclusion:
The conclusions are very simple and no statistical support is provided. If the intention of this article is a review, then it is necessary to show the experiments that other authors performed in order to reach these conclusions that you mentioned.
Author Contributions is missing
“Author ContributionsAuthor Contributions: For research articles with several authors, a short paragraph specifying their individual contributions must be provided. The following statements should be used “Conceptualization, X.X. and Y.Y.; methodology, X.X.; software, X.X.; validation, X.X., Y.Y. and Z.Z.; formal analysis, X.X.; investigation, X.X.; resources, X.X.; data curation, X.X.; writing—original draft preparation, X.X.; writing—review and editing, X.X.; visualization, X.X.; supervision, X.X.; project administration, X.X.; funding acquisition, Y.Y. All authors have read and agreed to the published version of the manuscript.” Please turn to the CRediT taxonomy for the term explanation. Authorship must be limited to those who have contributed substantially to the work reported.”
In my opinion, this report lacks scientific rigor, it seems an introduction of a thesis, which is different than a review, because in the second you need to show support to other authors' conclusions and then you make your own inferences.
Author Response
Manuscript ID: fermentation-1309224
Response to the comments of Reviewer 2
We are very much grateful for the time, sincere help, and patience of the Reviewer. We have modified the whole manuscript taking into account the comments of the Reviewer, which are addressed line-by-line as shown below. Replies come underneath in red.
Important remarks:
I strongly recommend for this article be revised by a native English speaker since some paragraphs are badly written.
Yes, we have linguistically revised the whole manuscript hoping that this version is better than the former.
Minor remarks:
Regarding this paragraph “To support this view, we synthesized the literature for the association between vitamin K and SARS-CoV-2 based on the findings reported from machine learning studies, molecular simulation, and human studies.”
What do you mean by synthesizing the literature?
We meant “summarizing” the literature. We have rephrased the sentence to make it clearer (line 35-41).
Figure 1:
In “Figure 1. Schematic illustration of common physiological functions of vitamin K, possible mechanisms of its deficiency in COVID-19, and associated adverse effects in deficient patients.”
This description is wrong since no mechanisms are shown to demonstrate deficiency. The figure is very uninformative and the colors are not used to guide the read and interpretation of the figure but are randomly chosen.
Yes, thank you very much. We agree with the reviewer; that figure was quite messy. Accordingly, we have re-drawn the figure, modified the color, added a detailed illustration to the caption of Figure 1. As the reviewer noted, the mechanism was not really addressed. Therefore, an additional figure (Figure 2) was included illustrating a speculation of the possible mechanism of vitamin K deficiency in COVID-19.
Figure 2:
In this case, the figure is difficult to follow it has 3 arrows misplaced and the function of the blue box surrounding an area is incomprehensible.
Thank you once again. We have modified the Figure: removed the blue box, placed the arrows correctly, and added an illustration to the caption. It has been changed as Figure 3 in the current version.
Figure 3:
The colors make the figure difficult for the reader, they are distractive because the reader is trying to understand if the colors have a meaning in describing the figure.
That’s right; the figure is a mix of randomly selected colors. We have modified it accordingly. It has been changed as Figure 5 in the current version.
Conclusion:
The conclusions are very simple and no statistical support is provided. If the intention of this article is a review, then it is necessary to show the experiments that other authors performed in order to reach these conclusions that you mentioned.
The reviewer’s comment is quite enlightening and correct. We have rewritten the conclusion to make it a focused summary of the basic themes of the manuscript.
Author Contributions is missing
“Author ContributionsAuthor Contributions: For research articles with several authors, a short paragraph specifying their individual contributions must be provided. The following statements should be used “Conceptualization, X.X. and Y.Y.; methodology, X.X.; software, X.X.; validation, X.X., Y.Y. and Z.Z.; formal analysis, X.X.; investigation, X.X.; resources, X.X.; data curation, X.X.; writing—original draft preparation, X.X.; writing—review and editing, X.X.; visualization, X.X.; supervision, X.X.; project administration, X.X.; funding acquisition, Y.Y. All authors have read and agreed to the published version of the manuscript.” Please turn to the CRediT taxonomy for the term explanation. Authorship must be limited to those who have contributed substantially to the work reported.”
Yes, thank you very much. We have completed the author contribution section.
In my opinion, this report lacks scientific rigor, it seems an introduction of a thesis, which is different than a review, because in the second you need to show support to other authors' conclusions and then you make your own inferences.
The author is entirely righteous. The relative novelty of COVID-19 has precluded the implementation of rigorous experimental studies, including RCTs, that investigate the anti-COVID-19 effects of various natural compounds with documented immunomodulatory activities such as those addressed in the current manuscript (fermented milk and bee honey). Therefore, shedding the light on the possible immune-enhancing roles that these products may play in vulnerability to COVID-19 and its severity may pave the way for conducting sound RCTs that would explore such potential. We have noted in many instances that the roles to be exerted by fermented milk and bee honey are speculated based on the bioactivities documented in the literature.
We hope that the manuscript has been satisfactorily modified and that the current version will be suitable for publication.
Best regards,
Corresponding author

Round 2
Reviewer 1 Report
The authors reply to the reviewers comments are adequate.
Author Response
Vitamin K in COVID-19—potential anti-COVID-19 properties of fermented milk fortified with bee honey as a natural source of vitamin K and probiotics.
Response to Comments of Reviewer 1
We thank Reviewer 1 for the time, effort, productive and insightful comments, which have made a real difference in the quality the current version of the manuscript. We hope that the revised version will be suitable for publication.
Best regards,
Corresponding author

Reviewer 2 Report
Review of the second version of the article:
Vitamin K in COVID-19—potential anti-COVID-19 properties of fermented milk fortified with bee honey as a natural source of vitamin K and probiotics.
Important remarks:
The article has improved, nevertheless, I think it is not ready yet for publication.
Figure 1: Still the arrangement of the figure is hard to understand, Are the three items related to the following 2 beige boxes??
Why do you arrange the information in 3 items, then in two boxes and then into several smaller boxes?
Figure 2: I recommend to increase the figure size, and font, also, in the legend you mentioned three main items. I recommend to include these points, 1, 2, and 3 in the figure so that it can be easier to follow.
The phrase “Vitamin K and its analogue, Kappadione, inhibited NSPs necessary for the replication of SARS- CoV-2 such as the main protease (Mpro/3CLpro) and RNA-dependent RNA polymerase (RdRp) in silico [63,64].” Is not true since in figure 4 just some loose molecule structures are depicted, without a particular order. I don’t understand the function of this figure, because it is not present to demonstrate anything.
If you mentioned this in the article, excuse me, but Is there any report comparing the intake of fermented milk fortified with bee honey as a palliative against the effect of vitamin K antagonists (VKA)? I think this is something that needs to be present in the discussion as well as the conclusions.
In general, the article is very speculative, mostly about the effect of the fermented milk fortified with bee honey. Perhaps changing the subtitles, from 6.3.1 to 6.3.6 into a more concise/strictly related with the discussion inside each of these subtitles could help avoid that.
Then at the conclusions, you could indicate that based on these premises in 6.3.X it can be inferred that supplementing with fermented milk fortified with bee honey could protect against COVID-19 infection.
Author Response
Vitamin K in COVID-19—potential anti-COVID-19 properties of fermented milk fortified with bee honey as a natural source of vitamin K and probiotics.
Response to Comments of Reviewer 2
We appreciate the Reviewer’s help improving our manuscript through such thoughtful reading and concerns for clarity as indicated by the provided comments. The comments are addressed line-by-line as shown below. Replies come underneath in red.
Figure 1: Still the arrangement of the figure is hard to understand, Are the three items related to the following 2 beige boxes??
Why do you arrange the information in 3 items, then in two boxes and then into several smaller boxes?
Yes, thank you. For clarity, we have modified the figure by including related elements in one single box.
Figure 2: I recommend to increase the figure size, and font, also, in the legend you mentioned three main items. I recommend to include these points, 1, 2, and 3 in the figure so that it can be easier to follow.
Yes, we have increased the size of the figure and the font. We have also included points (1, 2, and 3) in the figure.
The phrase “Vitamin K and its analogue, Kappadione, inhibited NSPs necessary for the replication of SARS- CoV-2 such as the main protease (Mpro/3CLpro) and RNA-dependent RNA polymerase (RdRp) in silico [63,64].” Is not true since in figure 4 just some loose molecule structures are depicted, without a particular order. I don’t understand the function of this figure, because it is not present to demonstrate anything.
The reviewer is right. Although that figure presents some loose molecule structures, the text is correct and more comprehensive. Therefore, we have removed that figure completely.
If you mentioned this in the article, excuse me, but Is there any report comparing the intake of fermented milk fortified with bee honey as a palliative against the effect of vitamin K antagonists (VKA)? I think this is something that needs to be present in the discussion as well as the conclusions.
Thank you for such a far-sighted comment. Based on this comment, we have searched PubMed and Google Scholar using terms “fermented milk and vitamin K antagonists”, but no relevant results were obtained from the search. So, we noted in the discussion that experimental evidence examining the protective effects of fermented milk or raw bee honey on the adverse effects of VKA is lacking, and research is invited to fill this gap (line 476-478).
In general, the article is very speculative, mostly about the effect of the fermented milk fortified with bee honey. Perhaps changing the subtitles, from 6.3.1 to 6.3.6 into a more concise/strictly related with the discussion inside each of these subtitles could help avoid that.
Then at the conclusions, you could indicate that based on these premises in 6.3.X it can be inferred that supplementing with fermented milk fortified with bee honey could protect against COVID-19 infection.
Yes, thank you very much. We have tried our best to change the subtitles, from 6.3.1 to 6.3.6. We have noted in the conclusion that based on these premises in Section 6.3 we speculated that the use of fermented milk-fortified with honey as a dietary supplement may protect against COVID-19.
We thank the reviewer once again for his/her sincere advice. We hope that the comments were properly handled and that the revised version will be suitable for publication.
Best regards,
Corresponding author

Round 3
Reviewer 2 Report
After a thorough review, I have no further questions or concerns.